# Exploratory Transcriptomic Profiling Reveals the Role of Gut Microbiota in Vascular Dementia

**DOI:** 10.3390/ijms24098091

**Published:** 2023-04-30

**Authors:** Navdeep Kaur, Geneva LaForce, Deepthi P. Mallela, Prasenjit Prasad Saha, Jennifer Buffa, Xinmin S. Li, Naseer Sangwan, Kasia Rothenberg, Weifei Zhu

**Affiliations:** 1Department of Cardiovascular & Metabolic Sciences, Cleveland Clinic, Cleveland, OH 44195, USA; kaurn7@ccf.org (N.K.); laforcg@ccf.org (G.L.); malleld@ccf.org (D.P.M.); sahap2@ccf.org (P.P.S.); buffaj@ccf.org (J.B.); lix2@ccf.org (X.S.L.); sangwan@ccf.org (N.S.); 2Microbial Sequencing & Analytics Resource (MSAAR) Facility, Shared Laboratory Resources (SLR), Lerner Research Institute, Cleveland Clinic, Cleveland, OH 44195, USA; 3Cleveland Clinic Lou Ruvo Center for Brain Health, Cleveland, OH 44195, USA; rothenk@ccf.org

**Keywords:** vascular dementia (VaD), choline, trimethylamine-N-oxide (TMAO), RNA-sequencing, neuroinflammation

## Abstract

Stroke is the second most common cause of cognitive impairment and dementia. Vascular dementia (VaD), a cognitive impairment following a stroke, is common and significantly impacts the quality of life. We recently demonstrated via gut microbe transplant studies that the gut microbe-dependent trimethylamine-N-oxide (TMAO) pathway impacts stroke severity, both infarct size and long-term cognitive outcomes. However, the molecular mechanisms that underly the role of the microbiome in VaD have not been explored in depth. To address this issue, we performed a comprehensive RNA-sequencing analysis to identify differentially expressed (DE) genes in the ischemic cerebral cortex of mouse brains at pre-stroke and post-stroke day 1 and day 3. A total of 4016, 3752 and 7861 DE genes were identified at pre-stroke and post-stroke day 1 and day 3, respectively. The Kyoto Encyclopedia of Genes and Genomes (KEGG) enrichment analysis indicated pathways of neurodegeneration in multiple diseases, chemokine signaling, calcium signaling, and IL-17 signaling as the key enriched pathways. Inflammatory response genes interleukin-1 beta (*Il-1β*), chemokines (C–X–C motif chemokine ligand 10 (*Cxcl10*), chemokine ligand 2 (*Ccl2*)), and immune system genes (S100 calcium binding protein 8 (*S100a8*), lipocalin-2 (*Lcn2*)) were among the most significantly upregulated genes. Hypocretin neuropeptide precursor (*Hcrt*), a neuropeptide, and transcription factors such as neuronal PAS domain protein 4 (*Npas4*), GATA binding protein 3 (*Gata3*), and paired box 7 (*Pax7*) were among the most significantly downregulated genes. In conclusion, our results indicate that higher plasma TMAO levels induce differential mRNA expression profiles in the ischemic brain tissue in our pre-clinical stroke model, and the predicted pathways provide the molecular basis for regulating the TMAO-enhanced neuroinflammatory response in the brain.

## 1. Introduction

Vascular dementia (VaD), a multifactorial disorder characterized by changes to memory, thinking, and behavior of an individual, is caused by damage to brain tissue due to decreased blood flow. Blockage of arteries due to blood clots, atherosclerosis, and autoimmune diseases results in diminished blood supply to the brain after a stroke [1]. VaD is the second most prevalent form of dementia, accounting for approximately 20% of all dementia cases [2]. Age, gender, genetics, environmental variables, pre-stroke cognitive level, number of past strokes, and gut microbiome contributions are the major risk factors for VaD [3,4,5]. The genetic factors include vascular encephalopathies such as cerebral autosomal dominant arteriopathy with subcortical infarcts and leukoencephalopathy, autosomal recessive arteriopathy with subcortical infarcts and leukoencephalopathy, and hereditary cerebral hemorrhage with amyloidosis [4]. Additionally, environmental factors, namely air pollution and lifestyle factors such as diet and exercise, have been shown to constitute important risk factors for VaD [3,5]. Nearly one-third of stroke survivors have significant cognitive impairment within the first few months after stroke onset [6].

Stroke induces both an inflammatory brain reaction and the body’s immunological response, and there is mounting evidence that many factors may interact and contribute to dementia [7]. The severity of brain damage, its location, and the number of individual cerebrovascular lesions are among the most significant pathogenic determinants of post-stroke VaD [8]. In response to neuronal injury, local and systemic generation of chemokines, cytokines, and reactive oxygen species trigger neuroinflammation in the brain [9]. Subsequently, partial impairment of the blood–brain barrier caused by inflammation allows immune cells, such as lymphocytes, to enter the brain, and microglia, to become activated in the brain, resulting in impaired brain function [10]. Although the exact mechanisms linking stroke to dementia are still unknown, a growing body of clinical research suggests that neuroinflammation is a key driver in the development of dementia post-stroke [11,12,13,14]. In addition, both stroke and dementia are influenced by glial activation and the production of pro-inflammatory cytokines, which might lead to the deposition of amyloid-β and phosphorylated tau as the key hallmark features of dementia [15].

The pathophysiology of VaD is complex. However, with the advent of next-generation sequencing, few studies have provided valuable insights into the underlying pathophysiology at the level of altered molecular pathways [16,17,18]. Recent studies have suggested a plausible role of gut microbial dysbiosis in the pathology of cerebrovascular diseases via the gut–brain axis [19,20]. Trimethylamine-N-oxide (TMAO), a gut microbial-dependent metabolite, is derived primarily from dietary choline which is metabolized into trimethylamine (TMA) through the action of gut microbiota. TMA is subsequently oxidized in the host liver to TMAO by flavin-monooxygenases [21,22,23]. Several recent studies have implicated the involvement of gut microbial-dependent TMAO in cerebrovascular disease risk in both humans and mice [24,25,26,27,28]. Additionally, a gut microbial transplant study by our group showed that the TMAO pathway affects stroke severity and cognitive outcomes. We found that dietary choline supplementation increases TMAO levels and causes long-term cognitive deficits following stroke as evidenced by Y-maze and Barnes maze [29].

Although clinical studies have confirmed the association between the microbiome-gut–brain axis and vascular cognitive impairment (cerebral small vessel disease and post-stroke cognitive impairment) [30,31,32], the molecular mechanisms of TMAO-enhanced VaD remain unknown. Studies to this point have been focused on the association between TMAO levels and a dementia phenotype, but very few reports have performed molecular investigations. Hence, it is crucial to understand the pathophysiology of VaD by identifying specific genes and related pathways through whole transcriptome analysis of relevant animal models.

In this study, we aimed to explore the interaction between VaD and TMAO levels at the genome-wide level. We used a Rose Bengal Cold Light pre-clinical model of stroke and analyzed the ischemic side of the cortex before stroke, and at post-stroke day 1 and day 3 using RNA-sequencing. Our results showed that TMAO-mediated changes in neuroinflammation and neurodegeneration processes precluded onset of VaD. Therefore, our findings offer a foundational understanding of the molecular pathways involved in VaD that will be important for developing effective treatment strategies.

## 2. Results

### 2.1. Elevated Plasma Trimethylamine-N-Oxide (TMAO) Levels and Higher Infarct Volume in an Established Cognitively Deficit Stroke Mouse Model

With our previously established Rose Bengal Cold Light stroke model, we first confirmed the effect of choline supplementation on the development of stroke. The mice were fed either a control (0.08% total choline) or choline (1%) diet for three weeks prior to stroke injury. At 24 h post-stroke, plasma TMAO levels and infarct volume were evaluated (Appendix A). As shown in Supplementary Figure 1B, the 1% choline diet resulted in a significant increase in plasma TMAO levels compared to the control diet. Furthermore, we observed higher infarct volume in the choline group compared to the control (Appendix A). In accordance with our previous findings, choline supplementation enhanced TMAO generation and stroke susceptibility in vivo resulting in long-term cognitive deficits post-stroke [29]. Our group also evaluated the long-term functional outcomes in the established stroke model using Barnes maze and Y-maze tests; we found that choline supplementation leads to long-term cognitive deficits post-stroke [29].

To investigate the molecular role of the gut microbial TMAO pathway on host susceptibility for VaD, we first induced stroke in mice using the Rose Bengal Cold Light stroke model. Additionally, plasma TMAO levels were measured at each time point (Appendix A). Plasma TMAO concentrations (mean ± SEM) ranged from 103.09 ± 46.10 to 163.42 ± 56.56 µM for the choline group, and 2.26 ± 0.68 to 3.98 ± 1.58 µM for the control group. Across all time points, mice in the choline group had substantially higher plasma TMAO levels than controls (*p* < 0.05, Mann–Whitney U test; Appendix A).

### 2.2. Ordination Analysis

We collected the ischemic hemisphere of the cerebral cortex at pre-stroke and post-stroke days 1 and 3 for RNA-sequencing. Correspondence analysis showed that biological replicates between the two groups (choline and control) clustered together (Figure 1A). The groups divided along the correspondence analysis component CA1 which accounted for 55–67.4% of the variance, whereas CCA1 separated the three time points, accounting for 18.4% of the variance. We observed significant differences between the choline and control groups for transcriptional changes (PERMANOVA r^2^ = 0.46, *p* < 0.001). Taken together, these results indicated that the effect of the treatment between choline and control group is greater than the time points.

### 2.3. Gene Expression Analysis

We performed differential gene expression analysis to define significantly differentially expressed (DE) genes between the choline and control groups. The read counts aligned to mouse reference genome (mm10) were analyzed in terms of fold change (FC) in gene expression. We discovered 4016 significantly DE genes at pre-stroke, 3752 on post-stroke day 1, and 7861 genes on post-stroke day 3. As shown in the scattered volcano plots (Figure 1B–D), genes that are outside the threshold lines of log transformed FC ≥ ±1.5 and −log10 *p*-value ≤ 0.05 are colored red. Statistically significant genes with FC < ±1.5 are shown in the color blue, whilst statistically insignificant genes with FC > ±1.5 are represented in green. There were slight changes in the expression levels of many genes in both directions, but only the genes with FC ≥ 1.5 and *p*-value ≤ 0.05 that met our screening criteria were reported here in this study. A total of five significantly DE genes were identified at pre-stroke, including three upregulated (*Cyp2a5*, *Wnt9b*, *Gfy*) and two (*Prl*, *Gh*) downregulated genes (Figure 1B). On post-stroke day 1, the total number of genes with DE at a set threshold for FC ≥ 1.5 and adjusted *p*-value ≤ 0.05 was 124, of which 116 were upregulated and 8 were downregulated, while at day 3, we found 252 DE genes, including 200 upregulated and 52 downregulated genes (Appendix A). The genes labelled on the volcano plots in black color represent some of the significant hits determined after multiple testing correction (Benjamini–Hochberg FDR correction). The orange-colored genes represent the ones with the highest FC (in both directions) at each of the tested time points. Overall, the key significantly upregulated genes based on FC values were *Cyp2a5*, *Tgm1*, and *Lcn2* (2.4, 4.57, and 4.26 folds, respectively), whereas downregulated genes were *Prl*, *Hcrt*, and *Pou4f2* (−4.35, −3.3, and −6.1 folds, respectively), at pre-, post-stroke day 1 and day 3, respectively (Figure 2B–D; Appendix A). In addition, we analyzed post-stroke day 1 and day 3 to find the genes that were overlapping, linking the time points with the progression of the neurovascular condition over time. We found 46 DEGs including 40 upregulated and 6 downregulated genes that were shared between day 1 and day 3 (Appendix A).

### 2.4. Functional Classification and Pathway Enrichment Analysis of Differentially Expressed (DE) Genes

To further explore the patterns of gene expression in choline versus control groups, a hierarchical clustering analysis was performed based on normalized z-score values of the identified DE genes (Figure 2). For all DE genes, the data demonstrated significant separation between the transcriptomes of the choline and control groups with minor change across the biological replicates (at set threshold of log FC > ±1.5; *p* < 0.05). The gene expression in each sample in both comparison groups is shown in Figure 2, displaying the top one hundred most significant genes selected based on the FC values and their role in neurodegenerative disease-related pathways. We observed distinct patterns of gene expression between choline and control groups for genes belonging to pathways of (1) neurodegeneration-multiple diseases (*Tubb6*, *Wnt9b*, *Cybb*, and *Tuba1c*, as marked in red color), which is an aggregation of pathways including Alzheimer’s disease, Huntington’s disease and Parkinson’s disease, (2) IL-17 signaling (*Mmp3*, *Cxcl10*, *Lcn2*, *S100a8*, *S100a9*, and *Il-1β*, as marked in green color), (3) MAPK signaling (*Flnc*, *Areg*, *Cd14*, and *Tnfrsf1a*, as marked in orange color), (4) calcium signaling (*Bdkrb2* and *Atp2a1*, as marked in pink color) and (5) PI3-Akt signaling (*Gh*, *Prl*, *Spp1*, *Tlr2*, and *Ntf5*, as marked in blue color) (Figure 2).

We then further determined the global function of genes upregulated or downregulated in the choline group by performing Gene Ontology (GO) enrichment analysis. We identified the top five significantly enriched biological processes, molecular functions, and cellular components based on the highest gene enrichment and with *p* < 0.05 (the false discovery rate with Benjamini–Hochberg correction) as shown in Appendix A. The most enriched biological processes such as biological regulation, regulation of cellular process, and response to stimulus were detected for the genes DE in choline group at post-stroke day 1 and day 3. Moreover, the molecular functions related to DE genes at the three tested time points included processes (1) signaling receptor regulator activity, (2) protein binding and molecular function regulator activity, and (3) signaling receptor binding and signaling receptor activity.

To uncover pathway connectivity, the DE genes with an expression change of at least onefold in the choline group compared to controls were submitted to the iPathway Guide and the Kyoto Encyclopedia of Genes and Genomes (KEGG) database, as described in the Material and Methods section. Genes perform their function at different levels (e.g., cell and organism levels), and KEGG is a database to study the molecular biology of these gene clusters. According to this analysis, the most enriched significant pathways at pre-stroke were those associated with neurodegeneration-multiple diseases (183 DE genes), Alzheimer’s disease (152 DE genes), Huntington’s disease (124 DE genes), Parkinson’s disease (118 DE genes), PI3K-AKT signaling (83 DE genes), and MAPK signaling (70 DE genes). (Figure 3A and Appendix A.) Additionally, on day 1 and day 3 post-stroke, some of the most enriched pathways included those for neurodegeneration-multiple diseases (147 and 270 DE genes, respectively), Alzheimer’s disease (120 and 227 DE genes, respectively), calcium signaling (67 and 111 DE genes, respectively), chemokine signaling (60 and 101 DE genes, respectively), and MAPK signaling (87 and 136 DE genes, respectively). (Figure 3B,C and Appendix A.)

The comparison of expression changes in DE genes at pre-, post-stroke day 1 and day 3 suggested that the direction of the change was mostly maintained, pointing to activation of similar pathways at different time points. Next, we investigated whether the expression of the DE genes identified in the present study correlated with TMAO levels, using Spearman’s correlation analysis (Figure 3D). We observed that the genes belonging to pathways of neurodegeneration-multiple diseases (*Cybb* and *Wnt9b*) were among the significantly positively correlated genes with TMAO level in the choline group. On the other hand, IL-17 signaling genes (*Il-1β*, *S100a8*, *S100a9*, and *Ccl2*) showed weak correlation with TMAO level in the choline group (Figure 3D).

### 2.5. Verification of Selected DE Genes

To further confirm the gene expression changes we identified by RNA-sequencing, we performed RT-qPCR analysis. It was interesting to note that several of the genes implicated in IL-17 signaling, a pathway that enhances neuroinflammatory response, overlapped at both day 1 and day 3 post-stroke. Therefore, we chose to validate the common differentially regulated IL-17 signaling genes (*Il-1β*, *S100a8*, *S100a9*, and *Ccl2*) as well as a few top significant hits (*Tgm1*, *Hcrt*) from the dataset (Figure 4). We observed that the expression of the chosen genes was consistent with the RNA-sequencing in terms of the direction of expression change at the tested time points. However, the differences in magnitude of expression were most likely due to the different algorithms applied for estimation of fold change in each approach.

## 3. Discussion

Vascular dementia (VaD) is a common neurological entity that encompasses all types of dementia and may affect up to one-third of stroke survivors. The risk is the highest within six months after stroke, and still persists for decades even after adjusting for established dementia risk factors [33,34]. Unlike physical disability after stroke, cognitive function often deteriorates over time before being diagnosed as VaD or Alzheimer’s disease (AD). Several studies have raised the possibility that the microbiome may be a major risk factor for VaD susceptibility [35,36]. Using a mouse model of early-onset Alzheimer’s disease, the amyloid precursor protein/presenilin 1 (APP/PS1) transgenic model transfer of the microbiota from conventional-raised mice to germ-free mice resulted in increased cerebral pathology, providing evidence that microbiota plays a key role in the development of neurodegenerative diseases [37]. The APP/PS1 transgenic mouse overexpresses mutant human amyloid precursor protein and presenilin 1 genes, both of which are directed to neurons in the central nervous system.

Previous studies from independent groups linked higher circulating TMAO with cognitive function decline and dementia in experimental models [26,28]. According to Gao et al. [26], increased plasma TMAO levels in APP/PS1 mice were associated with deteriorating cognitive performance and AD pathology. Another study reported that aging-induced microbial dysbiosis, with resultant higher TMAO levels, causes an inflammatory state contributing to cognitive decline [28]. In addition, patients with AD dementia and moderate cognitive impairment were reported to have increased TMAO levels in their blood and cerebrospinal fluid [21]. The high levels of TMAO seen in human cerebrospinal fluid indicate that liver-derived TMAO may penetrate the blood–brain barrier, although the exact process is unknown [38]. Another study showed that circulating TMAO levels increase with age in both humans and mice [39]. Furthermore, in the brain, TMAO was shown to impair antioxidant enzymes and accelerate brain aging via induction of neuronal senescence, synaptic damage, and inhibition of mTOR signaling, thus contributing to cognitive dysfunction [39]. Consequently, according to both experimental and clinical investigations, elevated levels of TMAO may be causally related to cognitive impairment.

In the present study, we induced stroke with the Rose Bengal Cold Light model in mice fed on choline and a control diet. We observed elevated circulating TMAO levels in the choline group compared to the control group. Consistent with our previous report [29], we observed that elevated TMAO levels were correlated with the higher infarct volume. To further investigate the potential mechanism underlying TMAO-mediated cognitive decline, we used RNA-sequencing to conduct an extensive transcriptome study in the cortex of the ischemic mouse brain. The results from this study provide insights into the regulatory processes underlying TMAO-mediated signaling in VaD development (Figure 5). TMAO participates in a number of biological processes and pathways connected to human disease. Increased platelet reactivity and thrombotic potential [40], changes in lipid and hormonal balance [41], and induction of inflammation by activating the NLRP3 inflammasome [42] are some of the TMAO-mediated processes. However, the complete repertoire of active molecules, networks, and pathways of differentially expressed genes in the process are not fully characterized. Our study, therefore, may serve to fill part of this gap of knowledge.

We identified *Il-1β*, *Ccl2*, *Cybb, Wnt9b,* as well as some pathway-specific genes such as *S100a8* and *S100a9* (IL-17 signaling) among the common pro-inflammatory and signaling molecules whose expression was affected by higher TMAO levels in post-stroke animals (Figure 3D). The production of cytokines, chemokines, and matrix-degrading enzymes ultimately leads to progressive neurodegeneration which is one of the hallmark features of dementia [43]. To further understand the tissue and cell-specific context of the observed transcriptional changes, we looked for an overlap in our DE genes with neuroinflammation and stroke-related transcriptome datasets. We found similarities between the genes in our dataset and previously reported astrocyte activation signatures (such as *Gfap* and *Serpina3n*), microglia priming genes (*Clec7a*, *Cst7*, *Cybb*, *Lgals3*, *Mmp12*, and *Spp1*), and genes common to inflammatory response (*Il-1β*, *Ccl4*, and *Cxcl10*) [44,45,46,47,48,49]. Inflammation has been reported to have both deleterious and beneficial effects in stroke [50]. However, immune cell recruitment and consequent increased production of inflammatory mediators (cytokines/chemokines), as well as blood–brain barrier rupture, can induce swelling and neuronal death [51]. Additionally, post-stroke neuroinflammation has been linked to poor cognitive function [52,53].

In addition to inflammatory genes, *Lcn2,* encoding lipocalin-2, was shown to be one of the most substantially elevated genes at post-stroke day 3. Lcn2 is a glycoprotein that is highly expressed in the central nervous system following injury and has a role in inflammation and immune system modulation [54]. According to current research, Lcn2 modulates neuroinflammation and impacts cellular activity in the central and peripheral nervous systems [55,56]. Moreover, increased Lcn2 expression after cerebral ischemia has been linked to glial activation, neuroinflammation, and blood–brain barrier disruption, all of which contribute to neuronal death [57]. It has also been reported to influence cytokine synthesis, such as tumor necrosis factor (Tnf), interleukin-6 (Il-6), and interleukin-1 beta (Il-1β), which are secreted by glia and are ultimately responsible for neuroinflammation [56]. Furthermore, Lcn2 has a role in controlling cellular responses by regulating iron homeostasis or activating inflammatory pathways [58]. Lcn2 binds to iron and transports it into the cell, resulting in neurotoxic levels of intracellular iron [59,60]. Increased release of Lcn2 by reactive astrocytes after intracellular iron accumulation contributes to neuronal degeneration [61,62]. In humans, *Lcn2* has previously been shown to distinguish VaD from other kinds of dementia such as Alzheimer’s disease dementia [63].

Another gene, *S100a8*, also known as the damage-associated molecular pattern molecule, was observed to be involved in the IL-17 signaling process [64]. S100A8/A9 typically appears as a heterodimer complex comprised of the calcium-binding proteins S100a8 and S100a9. This complex has been linked to the etiology of a variety of inflammatory disorders, including Alzheimer’s disease, traumatic brain injury, and stroke [65,66]. S100A8/A9 functions as an endogenous ligand of TLR4 and a receptor for advanced glycation end products, stimulating the production of pro-inflammatory cytokines in microglia and causing brain damage via ERK/NF-B and JNK/NF-B signaling pathways [67,68].

Overall, our findings show that a group of genes (such as *Il-1β*, *Ccl2*, *S100a8/9,* and *Lcn2*) associated with IL-17 signaling was linked to VaD pathogenesis. Moreover, the mechanisms related to the aforementioned genes may lead to subtle alterations in the neuro-inflammatory environment. This illustrates their potential to be promising therapeutic targets of the TMAO-mediated neuro-inflammatory pathway. To the best of our knowledge, this is the first report of transcriptomic profiling correlating TMAO with VaD pathology in the mouse brain examining three different time points including pre- and post-stroke. Because our RNA-sequencing data are based on bulk tissue, it may be difficult to discern the expression signal by cell type composition, limiting our capacity to detect the genes altered in a cell-specific manner. The blood–brain barrier plays an important role in both stroke and dementia. Research suggests that the blood–brain barrier at cerebral vasculature might cause microinfarcts or microbleeds, eventually leading to VaD [69]. Therefore, it would be interesting to investigate the effect of differentially expressed genes in the development of VaD via disruption of the blood–brain barrier. Cerebrospinal fluid remains in close contact with brain tissue while being separated from other tissues due to the blood–brain barrier. Analysis of TMAO levels in the cerebrospinal fluid might provide useful insights into the pathophysiology of central nervous system diseases including stroke and vascular dementia.

## 4. Material and Methods

### 4.1. Reagents

Reagents were purchased from Sigma (St. Louis, MO, USA) unless otherwise stated.

### 4.2. Animals Models and Diet

All stroke model studies followed RIGOR guidelines for effective translational stroke research [70,71,72]. The experiments were carried out using 14-weeks-old C57BL6/J female mice. The mice were housed under a 12 h light/dark cycle in temperature-controlled cages in an on-site animal facility. In vivo model used was a Rose Bengal Cold Light stroke model. Mice were fed the control and choline diets (N = 5 each), the contents of which have been previously documented [73,74,75,76], for the specified amount of time (usually three weeks). The base diet, also known as the “Control” diet (Tekled Envigo, TD130104), was employed and contained a chemically specified amount of total choline content (0.08%, *w*/*w*), which was independently confirmed by mass spectrometry, as previously mentioned [76]. The same base Control diet was utilized in another diet, which included an extra 1.00% (*w*/*w*) free choline (referred to as the “Choline” diet; Tekled Envigo, TD09041). For most studies, mice were subjected to the Rose Bengal Cold Light stroke model following established methods [77]. Rose Bengal was administered intraperitoneally to mice under general anesthesia at a dosage of 1 mg per 10 g of body weight. Rose Bengal solution was freshly prepared the day of usage at a concentration of 10 mg/mL. The anesthetized mice were then placed in a prone position. To restrict the illuminated area, a skull mask with a hole of 30mm^2^ was applied. Starting five minutes after the Rose Bengal injection, the target brain region—2 mm laterally and 2 mm posterior to the bregma on the left hemisphere—was illuminated for 15 min by a cold light source (KL1500, Schott, Mainz, Germany). The mice were then returned to clean cages after being placed on a heating pad until regaining consciousness. Brain samples were excised from the animals at pre-, post-stroke day 1 and day 3 and immediately stored at −80 °C until sequencing.

### 4.3. TMAO Measurement

Blood samples were collected from each animal, and plasma was separated and stored at −80 °C until further analysis. The levels of TMAO in the plasma were measured using stable isotope dilution high-performance liquid chromatography (HPLC) with online electrospray ionization tandem mass spectrometry using an AB SCIEX 5000 triple quadrupole mass spectrometer interfaced with a Shimadzu HPLC system equipped with silica column (4.6 × 250 mm, 5 µm, Luna Silica; Regis, Minneaplois, MN, USA) at a flow rate of 0.8 mL/min. Calibration curves were generated by using different concentrations of the standards, and were used to quantitate TMAO level.

### 4.4. 2,3,5-Triphenyl-Tetrazolium Chloride (TTC) Staining

TTC was dissolved in saline at a concentration of 2% and used for staining brain slices. Brain slices (2 mm each) were incubated in glass vials containing 20 mL of TTC solution at room temperature for 40 min. The vials were gently swirled once after 20 min. After incubation, the TTC solution was discarded, and the brain slices were fixed in 10% neutral buffered formalin at room temperature overnight. After fixation, color images of brain slices were obtained by directly scanning the slices on a flatbed fluorescence stereomicroscope (Leica MZ 16 FA, Leica Biosystems, Nußloch, Germany).

### 4.5. RNA Extraction and Quality Assessment

Total RNA was extracted from left brain hemisphere in mice using the RNeasy lipid tissue mini kit (#74804, Qiagen, Hilden, Germany) following manufacturer’s instructions. RNA was eluted in 50 µL of RNase free water and stored at −80 °C until further use. RNA samples from tissues were quantified using a Nanodrop spectrophotometer (Thermo Fisher Scientific, Waltham, MA, USA). A_260_:A_280_ and A_260_:A_230_ values of ~2.0 were accepted as pure for RNA. For samples used in sequencing, the integrity of RNA was determined using TapeStation (Agilent Technologies, Machelen, Belgium). RNA Integrity Numbers (RIN) were used to evaluate the integrity of RNA samples with RIN values > 7.0 considered as intact RNA and values < 7.0 considered as degraded RNA.

### 4.6. RNA-Sequencing

RNA-sequencing was performed on the left cerebral hemisphere (containing the lesion) of each mouse brain from the choline and control groups. Briefly, 1 μg of total RNA from each sample was used to construct libraries which included reverse transcription and adapter ligation to 3′ and 5′ end of the RNA molecules. The resulting cDNA libraries were PCR amplified and the purified libraries were quantified on Qubit fluorometer using dsDNA HS Assay kit (catalog #Q32851, Thermo Fisher Scientific, Waltham, MA, USA) and outsourced for sequencing to CC Genomics Core.

### 4.7. Reverse Transcription and Quantitative Real Time PCR (RT-qPCR)

The RNA-sequencing results were validated by RT-qPCR. For gene expression analysis, 500 ng of total RNA was converted into cDNA using the High-Capacity cDNA Reverse Transcription kit (#4374966, Thermo Fisher Scientific, Waltham, MA, USA) according to the manufacturer’s instructions. In brief, each 20 µL RT reaction contained 50 ng/µL of RNA, 25× dNTP mix, 50 U/μL MultiScribe™ reverse transcriptase, 10× reverse transcription buffer, 10× RT random primers, and 20 U/μL RNase inhibitor. The reactions were incubated in T100 thermal cycler (BIO-RAD, Hercules, CA, USA) at 25 °C for 10 min, 37 °C for 120 min and 85 °C for 5 min followed by hold at 4 °C. Reverse transcription products were then used for qPCR reactions for each gene carried out in duplicates using housekeeping gene *GAPDH* as an internal control, and each 20 μL reaction mixture included 2 μL of RT product at a concentration of 2 ng/µL, 10 μL of TaqMan Fast Advanced Master Mix (catalog #4444557, Thermo Fisher Scientific, Waltham, MA, USA), 1 µL of 20× TaqMan gene expression assay (Thermo Fisher Scientific, Waltham, MA, USA), and 7 µL of nuclease free water. The reactions were amplified in StepOne Plus Real-Time PCR System (Applied Biosystems, Waltham, MA, USA) using cycling conditions; 95 °C for 20 s, followed by 40 cycles of 95 °C for 1 s and 60 °C for 20 s. The gene expression assays used in this study were: *Tgm1* (Assay ID: Mm00498375_m1), *Mmp3* (Assay ID: Mm00440295_m1), *Hcrt* (Assay ID: Mm01964030_s1), *Cxcl10* (Assay ID: Mm00445235_m1), *Il-1β* (Assay ID: Mm00434228_m1), *Lcn2* (Assay ID: Mm01324470_m1), *S100a8* (Assay ID: Mm00496696_g1), *Casp8* (Assay ID: Mm01255716_m1), and *Tlr7* (Assay ID: Mm00446590_m1) with *Gapdh* (Assay ID: Mm99999915_g1) used as the reference control. Results were presented as log_2_-transformed fold change values calculated by 2^−ΔΔCt^ method [78].

### 4.8. Statistics

#### 4.8.1. Sequencing Data 

Raw reads from the FASTQ files were processed using TrimGalore! v0.4.1 (Babraham Bioinformatics) to remove low-quality bases and adapter sequences and all of the downstream analyses were based on the high-quality clean reads. The filtered reads were mapped to mouse reference genome (mm10) from the Genome Reference Consortium. After mapping the reads with STAR Aligner v2.5.3, the abundance of gene expression was measured according to FPKM values (fragments per kilobase of exon per million fragments mapped) using Cufflinks v2.2.1. Differential gene expression analysis was performed to identify differentially expressed genes (DEGs) between the choline and control groups using DEseq2 [79]. Ordination analysis was performed using canonical correspondence analysis (CA) with Pearson chi-squared measure. Subsequently, genes were subset to include only those whose fold change was greater than 1 and *p*-value < 0.05 after multiple testing corrections (Benjamini–Hochberg FDR correction).

#### 4.8.2. Gene Ontology (GO) and Kyoto Encyclopedia of Genes and Genomes (KEGG) Enrichment Analysis

Gene Ontology (GO) enrichment of the differentially expressed genes (DEGs) was performed using enrichGO and enrichKEGG functions implemented in the clusterProfiler package [80]. GO terms categorized the DEG functions into biological processes (BP), cellular components (CC), and molecular function (MF).

#### 4.8.3. qPCR Data

RT-qPCR data include the average of two technical replicates for each sample. Sample sizes were indicated in the individual figures and results are represented as mean ± standard error of mean (SEM). Post hoc comparisons were made using the Mann–Whitney Test. Spearman correlation analysis was used to measure the TMAO–gene expression correlation. All statistical analyses were performed using the R software package version 4.1.3, and *p*-values < 0.05 were considered statistically significant.

## 5. Conclusions

In conclusion, we believe that the discovered differentially expressed genes and pathways might act as the prospective indicators for VaD, and the detailed insights into the transcriptional response of mouse brain to TMAO levels further expand the understanding of the role of gut microbiota in VaD. However, the underlying mechanisms should be validated in further experiments, providing a foundation for feasible treatment strategies.

## Figures and Tables

**Figure 1 ijms-24-08091-f001:**
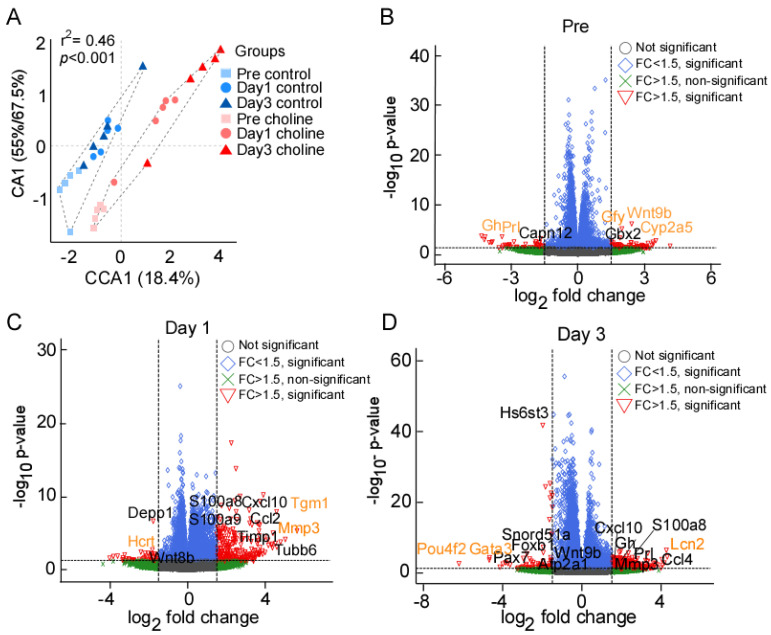
(**A**) Ordination (CA) plot separating the choline and control groups at three different time points; significance represented as chi-squared *p* < 0.05. (**B**–**D**) Volcano plot depiction of gene expression changes in choline and control groups at pre, day 1, and day 3 post-stroke, respectively. X-axis represents the log_2_ fold change for expressed genes, and y-axis represents significance of change in terms of *p*-value. The genes labelled in black color represent some of the significant hits determined after multiple testing correction (Benjamini–Hochberg FDR correction). The orange color represents the genes with highest fold change (in both directions) at each of the tested time points. The dotted lines represent the threshold cut off for fold change (FC) (1.5x fold change) and *p*-value (<0.05).

**Figure 2 ijms-24-08091-f002:**
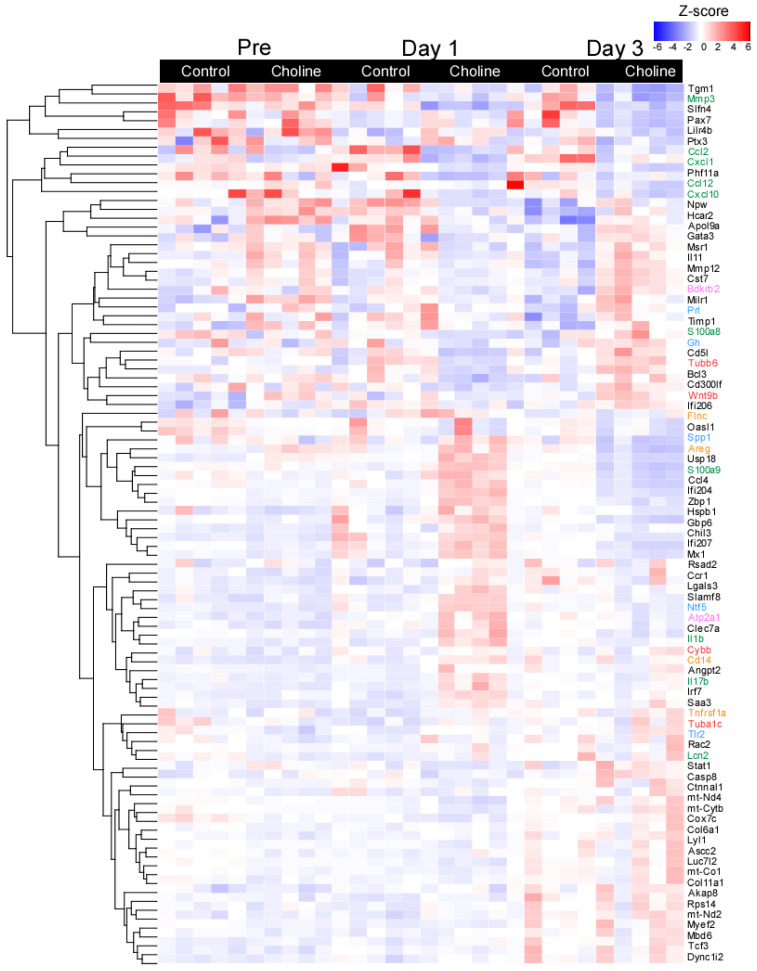
Heatmap of selected differentially expressed genes with fold change of ≥±1.5 and *p* ≤ 0.05. Each row represents a gene and the column represents a sample with colors indicating expression level based on normalized z-score. The red color indicates higher gene expression and blue represents lower gene expression. The line plots represent the expressed gene clusters as patterns.

**Figure 3 ijms-24-08091-f003:**
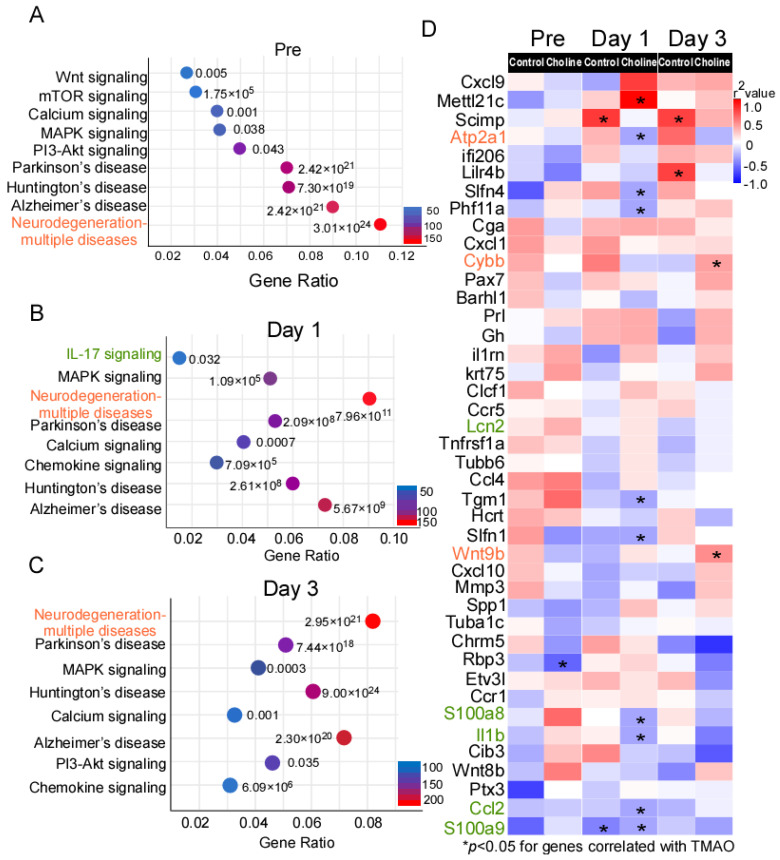
KEGG database analysis of the signaling pathway annotations for differentially expressed (DE) genes. (**A**–**C**) The most significantly enriched signaling pathways and corresponding *p*-values adjusted using Benjamini–Hochberg procedure, as well as number of differentially expressed (DE) genes in choline vs. controls. (**D**) Heatmap of Spearman correlations between gene expression and trimethylamine-N-oxide (TMAO) levels in choline and control groups. The corresponding significant *p*-values (*p* ≤ 0.05) are represented as asterisks and red color shows the higher correlation whereas blue represents weaker correlation. The genes in red belong to pathways of neurodegeneration-multiple diseases, green belongs to IL-17 signaling, and black are some of the top significant hits among the DE genes.

**Figure 4 ijms-24-08091-f004:**
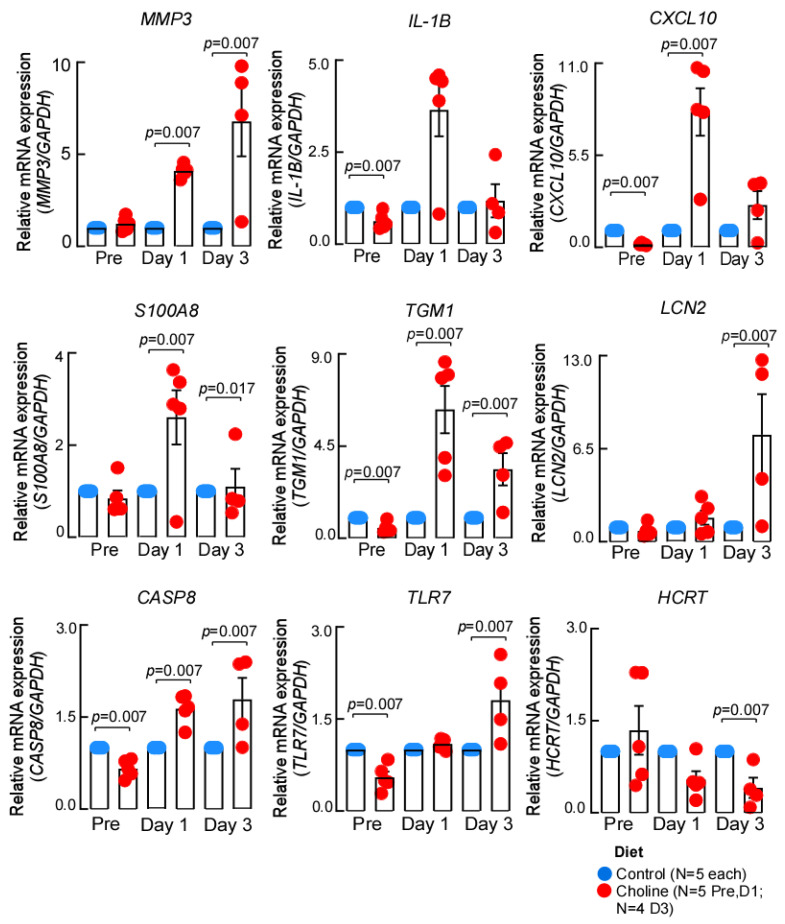
The expression patterns of genes validated by qPCR. The expression of each gene was normalized to *GAPDH*, using –ddCt method, data represented as mean ± SEM.

**Figure 5 ijms-24-08091-f005:**
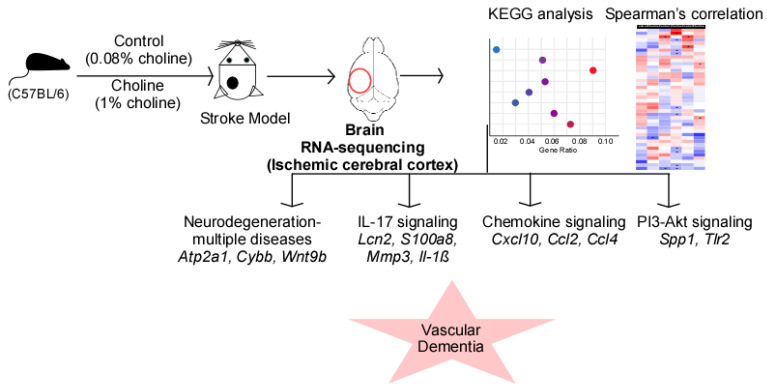
RNA-sequencing analysis revealed the role of trimethylamine-N-oxide (TMAO)-mediated signaling in vascular dementia. Elevated TMAO levels following choline diet result in altered mRNA expression profile in the ischemic cortex of the left brain hemisphere. The predicted signaling pathways shown in the scattered plot include pathways of neurodegeneration-multiple diseases, IL-17 signaling, chemokine signaling, and PI3-Akt signaling. The expression changes of the signaling molecules (italics) in these processes ultimately lead to the development of vascular dementia.

## Data Availability

All software and code used to analyze the current study are either open-source or commercially available (all included in Material and Methods section). The data set generated and or/analyzed during current study are available upon request from corresponding author, Weifei Zhu (zhuw@ccf.org).

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
