# Peer review of "Exploratory Transcriptomic Profiling Reveals the Role of Gut Microbiota in Vascular Dementia"

_ijms, 2023, doi:10.3390/ijms24098091_

Round 1
Reviewer 1 Report
Dear Dr.,
Title: Exploratory Transcriptomic Profiling Reveals the Role of Gut Microbiota in Vascular Dementia
Manuscript ID: ijms-2341488
Overall comments: Authors described in this manuscript: the role of gut microbiota function in vascular dementia with an exploration of transcriptomic profiling via comprehensive RNA-sequencing analytical techniques. The overall manuscript is written well and it has novelty in this field of research. The discussion and methodology section can be improved. The limitation is vascular dementia-related behavioural assessment is not incorporated; and TTC staining information. However, minor changes are required in this manuscript.
Specific comments:
1. If the author added additional behavioural assessment data on gut microbiota effect in vascular dementia is more valuable.
2. Experimental protocols approval number must be included.
3. TMAO measurement by HPLC graph can be incorporated as a supplementary file.
4. The following section's information needs to describe below the headings i.e., 4.1 Animals and General procedures and reagents. It is reflected as a mixed pattern.
5. TTC staining information is missing in this manuscript.
Minor comments
1. Keywords: Abbreviated terms should be avoided.
2. The introduction section is too lengthy. It can be rewritten in a concise manner.
*****
Author Response
Reviewer # 1
Overall comments: Authors described in this manuscript: the role of gut microbiota function in vascular dementia with an exploration of transcriptomic profiling via comprehensive RNA-sequencing analytical techniques. The overall manuscript is written well and it has novelty in this field of research. The discussion and methodology section can be improved. The limitation is vascular dementia-related behavioural assessment is not incorporated; and TTC staining information. However, minor changes are required in this manuscript.
Specific comments:
- If the author added additional behavioural assessment data on gut microbiota effect in vascular dementia is more valuable.
Author response: We really appreciate reviewer’s feedback. Sincerely, using various animal models and multiple behavioral testing to further study post-stroke vascular dementia is currently on-going in our group. However, this data is out of the scope of the current manuscript, which aims to report the transcriptomic profiling in the ischemic brain tissue earlier. In previous studies, we reported some cognitive testing and found that dietary choline supplementation increases TMAO levels and causes long-term cognitive deficits following stroke as evidenced by Y-maze and Barnes maze (Zhu et al. 2021). But we respectfully, prefer to report the more comprehensive study later.
- Experimental protocols approval number must be included.
Author response: As suggested by the reviewer, the protocol approval number has now been added under the heading Institutional Review Board Statement in the revised manuscript.
- TMAO measurement by HPLC graph can be incorporated as a supplementary file.
Author response: We added the graph as supplemental figure (Figure S2A) demonstrating cochromatography of two unique parent→daughter ion transitions for plasma TMAO (m/z 76.1®59.1) in control diet (blue line), choline diet (red line), and internal standard d9-trimethyl-TMAO (m/z 85.0®66.1)”.
- The following section's information needs to describe below the headings i.e., 4.1 Animals and General procedures and reagents. It is reflected as a mixed pattern.
Author response: As suggested by the reviewer, the information under heading 4.1 Animals and General procedures and reagents is now revised and separated into two different subheadings under materials and methods section as 4.1 Reagents and 4.2 Animal Models and Diet, in the revised manuscript.
- TTC staining information is missing in this manuscript.
Author response: The authors apologize for the missing information on TTC staining. This information is now added under the material and methods section, subheading 2,3,5-Triphenyl-tetrazolium chloride (TTC) staining.
Minor comments
- Keywords: Abbreviated terms should be avoided.
Author response: Thank you for pointing this out. We changed the abbreviated keyword “TMAO” to its full name “Trimethylamine-N-oxide.”
- The introduction section is too lengthy. It can be rewritten in a concise manner.
Author response: We appreciate the feedback from the reviewer. With regard to length of the introduction, we reviewed the introduction section carefully to remove the redundancies, in the revised manuscript.
Please also see the attachment.
Reviewer 2 Report
Comments to the Author
Journal name: International Journal of Molecular Sciences
The submitted article "Exploratory Transcriptomic Profiling Reveals the Role of Gut Microbiota in Vascular Dementia" is well written, structured and organized with original aspects concerning the role of the microbiome in vascular dementia at molecular levels. The subject is extremely relevant for the scientific community and society as a whole. The references are current and the figures are quite illustrative. The article should certainly be published after minor revisions. However, I have some suggestions which should be considered:
1) Some abbreviations need to be revised, such as BBB, CSF, etc.
2) In the section: "Conclusions", it would be convenient to add a statement in which the authors give their opinion on the perspective of this topic based on the results discussed in the manuscript.
3) I recommend the addition of a summary figure at the end of this interesting study that shows the molecular mechanisms underlying TMAO-mediated signaling in the development of vascular dementia including the genes linked to this pathogenesis.
Author Response
Reviewer # 2
The submitted article "Exploratory Transcriptomic Profiling Reveals the Role of Gut Microbiota in Vascular Dementia" is well written, structured and organized with original aspects concerning the role of the microbiome in vascular dementia at molecular levels. The subject is extremely relevant for the scientific community and society as a whole. The references are current and the figures are quite illustrative. The article should certainly be published after minor revisions. However, I have some suggestions which should be considered:
- Some abbreviations need to be revised, such as BBB, CSF, etc.
Author response: We agree with the reviewer’s suggestion. Accordingly, throughout the manuscript we revised the abbreviations at their first appearance.
- In the section: "Conclusions", it would be convenient to add a statement in which the authors give their opinion on the perspective of this topic based on the results discussed in the manuscript.
Author response: As per reviewer’s suggestion, we added a separate conclusions section within the revised manuscript, stating our opinion on the results discussed in this study.
- I recommend the addition of a summary figure at the end of this interesting study that shows the molecular mechanisms underlying TMAO-mediated signaling in the development of vascular dementia including the genes linked to this pathogenesis.
Author response: We thank the reviewer for this assessment. We added a summary figure 5 in the discussion section (paragraph 3, sentence 5) showing molecular mechanisms underlying vascular dementia, in the revised manuscript.
Please also see the attachment.
Reviewer 3 Report
My suggestions:
1. In the introduction, I would add a few examples of genetic and environmental factors, which may impact vascular dementia.
2. In part Methods, at stroke model, how many control and Choline diet? I would also describe the diets a little bit more in detail.
3. I would add a chapter from the introduction to the discussion. The part, where the authors described the examples of mouse models on gut microbiota in other diseases would suit better in the discussion. In the introduction, I would mention such mouse models only briefly.
4. I think, Figure S1 would suit well to the Methods chapter instead the supplement.]
5. In Figure 1, what do the black and orange colors mean? The authors may mention it.
6. Did the authors analyze the TMAO in the CSF of the mice? It would also be an interesting study or a great follow-up study.
7. I would add a figure, on how the differently expressed genes may impact Vascular dementia in mice through blood brain barrier.
Author Response
Reviewer # 3
- In the introduction, I would add a few examples of genetic and environmental factors, which may impact vascular dementia.
Author response: Thank you for this suggestion. We added a few sentences providing examples of genetic and environmental factors in the first paragraph, sentences 5 and 6, in the introduction section of the revised manuscript.
- In part Methods, at stroke model, how many control and Choline diet? I would also describe the diets a little bit more in detail.
Author response: As per reviewer’s suggestion, the number of animals (N=5) in each diet group have been added under Materials and Methods section, heading 4.2 Animal Models and Diet, sentence 4. Additionally, the control and choline diet are now explained more under Materials and Methods section, heading 4.2 Animal Models and Diet, sentences 5-6.
- I would add a chapter from the introduction to the discussion. The part, where the authors described the examples of mouse models on gut microbiota in other diseases would suit better in the discussion. In the introduction, I would mention such mouse models only briefly.
Author response: We moved the suggested content from introduction to the discussion section paragraph 1.
- I think, Figure S1 would suit well to the Methods chapter instead the supplement.
Author response: We wish to note that we performed this set of experiment for this manuscript, not just to demonstrate the model or method, but also to confirm the effect of TMAO on stroke severity. We respectfully, prefer it as supplementary figure.
- In Figure 1, what do the black and orange colors mean? The authors may mention it.
Author response: As per reviewer’s suggestion the information about the color coding in Figure 1 is now added in the Figure Legend. Additionally, the information is added in the text under Results section, heading 2.3 Gene Expression Analysis, sentences 9-10.
- Did the authors analyze the TMAO in the CSF of the mice? It would also be an interesting study or a great follow-up study.
Author response: Thank you for this suggestion. In current study, TMAO levels in CSF were not analyzed. We agree with the reviewer that it would be very interesting to explore this aspect as follow-up study. We added this point into the last paragraph, sentence 8, in the discussion section of the revised manuscript.
- I would add a figure, on how the differently expressed genes may impact Vascular dementia in mice through blood brain barrier.
Author response: We are not able to add this figure since we did not investigate blood brain barrier in the current manuscript, and we need to return the revised version back to journal within 10 days. However, we agree with the reviewer’s suggestion, so we added the point that this will be an interesting follow-up study in the last paragraph, sentences 6-9, in the discussion section of the revised manuscript.
Please also see the attachment.
Round 2
Reviewer 3 Report
I think the authors fulfilled the suggestions well, thank you.